# The Large Margin Mechanism
# for Differentially Private Maximization

**Kamalika Chaudhuri**
UC San Diego
La Jolla, CA
kamalika@cs.ucsd.edu

**Daniel Hsu**
Columbia University
New York, NY
djhsu@cs.columbia.edu

**Shuang Song**
UC San Diego
La Jolla, CA
shs037@eng.ucsd.edu

## Abstract

A basic problem in the design of privacy-preserving algorithms is the *private maximization problem*: the goal is to pick an item from a universe that (approximately) maximizes a data-dependent function, all under the constraint of differential privacy. This problem has been used as a sub-routine in many privacy-preserving algorithms for statistics and machine learning.

Previous algorithms for this problem are either range-dependent—i.e., their utility diminishes with the size of the universe—or only apply to very restricted function classes. This work provides the first general purpose, range-independent algorithm for private maximization that guarantees approximate differential privacy. Its applicability is demonstrated on two fundamental tasks in data mining and machine learning.

## 1  Introduction

Differential privacy [17] is a cryptographically motivated definition of privacy that has recently gained significant attention in the data mining and machine learning communities. An algorithm for processing sensitive data enforces differential privacy by ensuring that the likelihood of any outcome does not change by much when a single individual's private data changes. Privacy is typically guaranteed by adding noise either to the sensitive data, or to the output of an algorithm that processes the sensitive data. For many machine learning tasks, this leads to a corresponding degradation in accuracy or utility. Thus a central challenge in differentially private learning is to design algorithms with better tradeoffs between privacy and utility for a wide variety of statistics and machine learning tasks.

In this paper, we study the *private maximization problem*, a fundamental problem that arises while designing privacy-preserving algorithms for a number of statistical and machine learning applications. We are given a sensitive dataset $D \subseteq \mathcal{X}^n$ comprised of records from $n$ individuals. We are also given a data-dependent objective function $f : \mathcal{U} \times \mathcal{X}^n \to \mathbb{R}$, where $\mathcal{U}$ is a universe of $K$ items to choose from, and $f(i, \cdot)$ is $(1/n)$-Lipschitz for all $i \in \mathcal{U}$. That is, $|f(i, D') - f(i, D'')| \leq 1/n$ for all $i$ and for any $D', D'' \in \mathcal{X}^n$ differing in just one individual's entry. Always selecting an item that exactly maximizes $f(\cdot, D)$ is generally non-private, so the goal is to select, in a differentially private manner, an item $i \in \mathcal{U}$ with as high an objective $f(i, D)$ as possible. This is a very general algorithmic problem that arises in many applications, include private PAC learning [25] (choosing the most accurate classifier), private decision tree induction [21] (choosing the most informative split), private frequent itemset mining [5] (choosing the most frequent itemset), private validation [12] (choosing the best tuning parameter), and private multiple hypothesis testing [32] (choosing the most likely hypothesis).

The most common algorithms for this problem are the *exponential mechanism* [28], and a computationally efficient alternative from [5], which we call the *max-of-Laplaces mechanism*. These

algorithms are general—they do not require any additional conditions on $f$ to succeed—and hence have been widely applied. However, a major limitation of both algorithms is that their utility suffers from an explicit *range-dependence*: the utility deteriorates with increasing universe size. The range-dependence persists even when there is a single clear maximizer of $f(\cdot, D)$, or a few near maximizers, and even when the maximizer remains the same after changing the entries of a large number of individuals in the data. Getting around range-dependence has therefore been a goal for designing algorithms for this problem.

This problem has also been addressed by recent algorithms of [31, 3], who provide algorithms that are range-independent and satisfy approximate differential privacy, a relaxed version of differential privacy. However, none of these algorithms is general; they explicitly fail unless additional special conditions on $f$ hold. For example, the algorithm from [31] provides a range-independent result only when there is a single clear maximizer $i^*$ such that $f(i^*, D)$ is greater than the second highest value by some margin; the algorithm from [3] also has restrictive conditions that limit its applicability (see Section 2.2). Thus, a challenge is to develop a private maximization algorithm that is both range-independent and free of additional conditions; this is necessary to ensure that an algorithm is widely applicable and provides good utility when the universe size is large.

In this work, we provide the first such general purpose range-independent private maximization algorithm. Our algorithm is based on two key insights. The first is that private maximization is easier when there is a small set of near maximizing items $j \in \mathcal{U}$ for which $f(j, D)$ is close to the maximum value $\max_{i \in \mathcal{U}} f(i, D)$. A plausible algorithm based on this insight is to first find a set of near maximizers, and then run the exponential mechanism on this set. However, finding this set directly in a differentially private manner is very challenging. Our second insight is that only the number $\ell$ of near maximizers needs to be found in a differentially private manner—a task that is considerably easier. Provided there is a margin between the maximum value and the $(\ell + 1)$-th maximum value of $f(i, D)$, running the exponential mechanism on the items with the top $\ell$ values of $f(i, D)$ results in approximate differential privacy as well as good utility.

Our algorithm, which we call the *large margin mechanism*, automatically exploits large margins when they exist to simultaneously (i) satisfy approximate differential privacy (Theorem 2), as well as (ii) provide a utility guarantee that depends (logarithmically) only on the number of near maximizers, rather than the universe size (Theorem 3). We complement our algorithm with a lower bound, showing that the utility of any approximate differentially private algorithm must deteriorate with the number of near maximizers (Theorem 1). A consequence of our lower bound is that range-independence cannot be achieved with pure differential privacy (Proposition 1), which justifies our relaxation to approximate differential privacy.

Finally, we show the applicability of our algorithm to two problems from data mining and machine learning: frequent itemset mining and private PAC learning. For the first problem, an application of our method gives the first algorithm for frequent itemset mining that simultaneously guarantees approximate differential privacy and utility independent of the itemset universe size. For the second problem, our algorithm achieves tight sample complexity bounds for private PAC learning analogous to the shell bounds of [26] for non-private learning.

## 2 Background

This section reviews differential privacy and introduces the private maximization problem.

### 2.1 Definitions of Differential Privacy and Private Maximization

For the rest of the paper, we consider randomized algorithms $\mathcal{A} : \mathcal{X}^n \to \Delta(\mathcal{S})$ that take as input datasets $D \in \mathcal{X}^n$ comprised of records from $n$ individuals, and output values in a range $\mathcal{S}$. Two datasets $D, D' \in \mathcal{X}^n$ are said to be *neighbors* if they differ in a single individual's entry. A function $\phi : \mathcal{X}^n \to \mathbb{R}$ is $L$-Lipschitz if $|\phi(D) - \phi(D')| \leq L$ for all neighbors $D, D' \in \mathcal{X}^n$.

The following definitions of (approximate) differential privacy are from [17] and [20].

**Definition 1** (Differential Privacy)**.** A randomized algorithm $\mathcal{A} : \mathcal{X}^n \to \Delta(\mathcal{S})$ is said to be $(\alpha, \delta)$-*approximate differentially private* if, for all neighbors $D, D' \in \mathcal{X}^n$ and all $S \subseteq \mathcal{S}$,

$$\Pr(\mathcal{A}(D) \in S) \leq e^{\alpha} \Pr(\mathcal{A}(D') \in S) + \delta.$$

The algorithm $\mathcal{A}$ is $\alpha$-*differentially private* if it is $(\alpha, 0)$-approximate differentially private.

Smaller values of the privacy parameters $\alpha > 0$ and $\delta \in [0, 1]$ imply stronger guarantees of privacy.

**Definition 2** (Private Maximization)**.** In the *private maximization problem*, a sensitive dataset $D \subseteq \mathcal{X}^n$ comprised of records from $n$ individuals is given as input; there is also a universe $\mathcal{U} := \{1, \ldots, K\}$ of $K$ items, and a function $f : \mathcal{U} \times \mathcal{X}^n \to \mathbb{R}$ such that $f(i, \cdot)$ is $(1/n)$-Lipschitz for all $i \in \mathcal{U}$. The goal is to return an item $i \in \mathcal{U}$ such that $f(i, D)$ is as large as possible while satisfying (approximate) differential privacy.

Always returning the exact maximizer of $f(\cdot, D)$ is non-private, as changing a single individuals' private values can potentially change the maximizer. Our goal is to design a randomized algorithm that outputs an approximate maximizer with high probability. (We loosely refer to the expected $f(\cdot, D)$ value of the chosen item as the utility of the algorithm.)

Note that this problem is different from private release of the *maximum value* of $f(\cdot, D)$; a solution for the latter is easily obtained by adding Laplace noise with standard deviation $O(1/(\alpha n))$ to $\max_{i \in \mathcal{U}} f(i, D)$ [17]. Privately returning a nearly maximizing item itself is much more challenging.

Private maximization is a core problem in the design of differentially private algorithms, and arises in numerous statistical and machine learning tasks. The examples of frequent itemset mining and PAC learning are discussed in Sections 4.1 and 4.2.

## 2.2 Previous Algorithms for Private Maximization

The standard algorithm for private maximization is the *exponential mechanism* [28]. Given a privacy parameter $\alpha > 0$, the exponential mechanism randomly draws an item $i \in U$ with probability $p_i \propto e^{n\alpha f(i,D)/2}$; this guarantees $\alpha$-differential privacy. While the exponential mechanism is widely used because of its generality, a major limitation is its *range-dependence*—i.e., its utility diminishes with the universe size $K$. To be more precise, consider the following example where $\mathcal{X} := \mathcal{U} = [K]$ and

$$f(i, D) := \frac{1}{n} \left| \{ j \in [n] : D_j \geq i \} \right| \tag{1}$$

(where $D_j$ is the $j$-th entry in the dataset $D$). When $D = (1, 1, \ldots, 1)$, there is a clear maximizer $i^* = 1$, which only changes when the entries of at least $n/2$ individuals in $D$ change. It stands to reason that any algorithm should report $i = 1$ in this case with high probability. However, the exponential mechanism outputs $i = 1$ only with probability $e^{n\alpha/2}/(K - 1 + e^{n\alpha/2})$, which is small unless $n = \Omega(\log(K)/\alpha)$. This implies that the utility of the exponential mechanism deteriorates with $K$.

Another general purpose algorithm is the *max-of-Laplaces* mechanism from [5]. Unfortunately, this algorithm is also range-dependent. Indeed, our first observation is that all $\alpha$-differentially private algorithms that succeed on a wide class of private maximization problems share this same drawback.

**Proposition 1** (Lower bound for differential privacy)**.** *Let $\mathcal{A}$ be any $\alpha$-differentially private algorithm for private maximization, $\alpha \in (0, 1)$, and $n \geq 2$. There exists a domain $\mathcal{X}$, a function $f : \mathcal{U} \times \mathcal{X}^n \to \mathbb{R}$ such that $f(i, \cdot)$ is $(1/n)$-Lipschitz for all $i \in \mathcal{U}$, and a dataset $D \in \mathcal{X}^n$ such that:*

$$\Pr \left( f(\mathcal{A}(D), D) > \max_{i \in \mathcal{U}} f(i, D) - \frac{\log \frac{K-1}{2}}{\alpha n} \right) < \frac{1}{2}.$$

We remark that results similar to Proposition 1 have appeared in [23, 2, 10, 11, 7]; we simply reframe those results here in the context of private maximization.

Proposition 1 implies that in order to remove range-dependence, we need to relax the privacy notion. We consider a relaxation of the privacy constraint to $(\alpha, \delta)$-approximate differential privacy with $\delta > 0$.

The approximate differentially private algorithm from [31] applies in the case where there is a single clear maximizer whose value is much larger than that of the rest. This algorithm adds Laplace noise with standard deviation $O(1/(\alpha n))$ to the difference between the largest and the second-largest values of $f(\cdot, D)$, and outputs the maximizer if this noisy difference is larger than $O(\log(1/\delta)/(\alpha n))$;

otherwise, it outputs `Fail`. Although this solution has high utility for the example in (1) with $D = (1, 1, \ldots, 1)$, it fails even when there is a single additional item $j \in \mathcal{U}$ with $f(j, D)$ close to the maximum value; for instance, $D = (2, 2, \ldots, 2)$.

[3] provides an approximate differentially private algorithm that applies when $f$ satisfies a condition called *$\ell$-bounded growth*. This condition entails the following: first, for any $i \in \mathcal{U}$, adding a single individual to any dataset $D$ can either keep $f(i, D)$ constant, or increase it by $1/n$; and second, $f(i, D)$ can only increase in this case for at most $\ell$ items $i \in \mathcal{U}$. The utility of this algorithm depends only on $\log \ell$, rather than $\log K$. In contrast, our algorithm does not require the first condition. Furthermore, to ensure that our algorithm only depends on $\log \ell$, it suffices that there only be $\leq \ell$ near maximizers, which is substantially less restrictive than the $\ell$-bounded growth condition.

As mentioned earlier, we avoid range-dependence with an algorithm that finds and optimizes over *near maximizers* of $f(\cdot, D)$. We next specify what we mean by near maximizers using a notion of *margin*.

## 3 The Large Margin Mechanism

We now our new algorithm for private maximization, called the *large margin mechanism*, along with its privacy and utility guarantees.

### 3.1 Margins

We first introduce the notion of *margin* on which our algorithm is based. Given an instance of the private maximization problem and a positive integer $\ell \in \mathbb{N}$, let $f^{(\ell)}(D)$ denote the $\ell$-th highest value of $f(\cdot, D)$. We adopt the convention that $f^{(K+1)}(D) = -\infty$.

**Condition 1** (($\ell, \gamma$)-margin condition). For any $\ell \in \mathbb{N}$ and $\gamma > 0$, we say a dataset $D \in \mathcal{X}^n$ satisfies the ($\ell, \gamma$)-*margin condition* if

$$f^{(\ell+1)}(D) < f^{(1)}(D) - \gamma$$

(*i.e.*, there are at most $\ell$ items within $\gamma$ of the top item according to $f(\cdot, D)$).[1]

By convention, every dataset satisfies the $(K, \gamma)$-margin condition. Intuitively, a $(\ell, \gamma)$-margin condition with a relatively large $\gamma$ implies that there are $\leq \ell$ near maximizers, so the private maximization problem is easier when $D$ satisfies an $(\ell, \gamma)$-margin condition with small $\ell$.

How large should $\gamma$ be for a given $\ell$? The following lower bound suggests that in order to have $n = O(\log(\ell)/\alpha)$, we need $\gamma$ to be roughly $\log(\ell)/(\alpha n)$.

**Theorem 1** (Lower bound for approximate differential privacy). *Fix any $\alpha \in (0, 1)$, $\ell > 1$, and $\delta \in [0, (1 - \exp(-\alpha))/(2(\ell - 1))]$; and assume $n \geq 2$. Let $\mathcal{A}$ be any $(\alpha, \delta)$-approximate differentially private algorithm, and $\gamma := \min\{1/2, \log((\ell - 1)/2)/(n\alpha)\}$. There exists a domain $\mathcal{X}$, a function $f : \mathcal{U} \times \mathcal{X}^n \to \mathbb{R}$ such that $f(i, \cdot)$ is $(1/n)$-Lipschitz for all $i \in \mathcal{U}$, and a dataset $D \in \mathcal{X}^n$ such that:*

1. *$D$ satisfies the $(\ell, \gamma)$-margin condition.*

2. $\Pr\left( f(\mathcal{A}(D), D) > f^{(1)}(D) - \gamma \right) < \dfrac{1}{2}.$

A consequence of Theorem 1 is that complete range-independence for all $(1/n)$-Lipschitz functions $f$ is not possible, even with approximate differential privacy. For instance, if $D$ satisfies an $(\ell, \Omega(\log(\ell)/(\alpha n)))$-margin condition only when $\ell = \Omega(K)$, then $n$ must be $\Omega(\log(K)/\alpha)$ in order for an approximate differentially private algorithm to be useful.

### 3.2 Algorithm

The lower bound in Theorem 1 suggests the following algorithm. First, privately determine a pair $(\ell, \gamma)$, with $\ell$ is as small as possible and $\gamma = \Omega(\log(\ell)/(\alpha n))$, such that $D$ satisfies the $(\ell, \gamma)$-margin

**Algorithm 1** The large margin mechanism LMM($\alpha, \delta, D$)

---

**input** Privacy parameters $\alpha > 0$ and $\delta \in (0, 1)$, database $D \in \mathcal{X}^n$.
**output** Item $I \in \mathcal{U}$.
  1: For each $r = 1, 2, \dots, K$, let

$$t^{(r)} := \frac{6}{n}\left(1 + \frac{\ln(3r/\delta)}{\alpha}\right) = O\left(\frac{1}{n} + \frac{1}{n\alpha}\log\frac{r}{\delta}\right),$$

$$T^{(r)} := \frac{3}{n\alpha}\ln\frac{3}{2\delta} + \frac{6}{n\alpha}\ln\frac{3}{\delta} + \frac{12}{n\alpha}\ln\frac{3r(r+1)}{\delta} + t^{(r)} = O\left(\frac{1}{n} + \frac{1}{n\alpha}\log\frac{r}{\delta}\right).$$

  2: Draw $Z \sim \text{Lap}(3/\alpha)$.
  3: Let $m := f^{(1)}(D) + Z/n$. {Estimate of max value.}
  4: Draw $G \sim \text{Lap}(6/\alpha)$ and $Z_1, Z_2, \dots, Z_{K-1} \overset{\text{iid}}{\sim} \text{Lap}(12/\alpha)$.
  5: Let $\ell := 1$. {Adaptively determine value $\ell$ such that $D$ satisfies $(\ell, t^{(\ell)})$-margin condition.}
  6: **while** $\ell < K$ **do**
  7:    **if** $m - f^{(\ell+1)}(D) > (Z_\ell + G)/n + T^{(\ell)}$ **then**
  8:      Break out of while-loop with current value of $\ell$.
  9:    **else**
 10:      Let $\ell := \ell + 1$.
 11:    **end if**
 12: **end while**
 13: Let $\mathcal{U}_\ell$ be the set of $\ell$ items in $\mathcal{U}$ with highest $f(i, D)$ value (ties broken arbitrarily).
 14: Draw $I \sim \boldsymbol{p}$ where $p_i \propto \mathbb{1}\{i \in \mathcal{U}_\ell\}\exp(n\alpha f(i, D)/6)$. {Exponential mechanism on top $\ell$ items.}
 15: **return** $I$.

---

condition. Then, run the exponential mechanism on the set $\mathcal{U}_\ell \subseteq \mathcal{U}$ of items with the $\ell$ highest $f(\cdot, D)$ values. This sounds rather natural and simple, but a knee-jerk reaction to this approach is that the set $\mathcal{U}_\ell$ itself depends on the sensitive dataset $D$, and it may have *high sensitivity* in the sense that membership of many items in $\mathcal{U}_\ell$ can change when a single individual's private value is changed. Thus differentially private computation of $\mathcal{U}_\ell$ appears challenging.

It turns out we do not need to guarantee the privacy of the set $\mathcal{U}_\ell$, but rather just of a valid $(\ell, \gamma)$ pair. This is essentially because when $D$ satisfies the $(\ell, \gamma)$-margin condition, the probability that the exponential mechanism picks an item $i$ that occurs in $\mathcal{U}_\ell$ when the sensitive dataset is $D$ but not in $\mathcal{U}_\ell$ when the sensitive dataset is its neighbor $D'$ is very small.

Moreover, we can find such a valid $(\ell, \gamma)$ pair using a differentially private search procedure based on the *sparse vector technique* [22]. Combining these ideas gives a general (and adaptive) algorithm whose loss of utility due to privacy is only $O(\log(\ell/\delta)/\alpha n)$ when the dataset satisfies a $(\ell, O(\log(\ell/\delta)/(\alpha n)))$-margin condition. We call this general algorithm the *large margin mechanism* (Algorithm 1), or LMM for short.

### 3.3 Privacy and Utility Guarantees

We first show that LMM satisfies approximate differential privacy.

**Theorem 2** (Privacy guarantee)**.** LMM($\alpha, \delta, \cdot$) *satisfies* $(\alpha, \delta)$-*approximate differential privacy.*

The proof of Theorem 2 is in Appendix A.1. The following theorem, proved in Appendix A.2, provides a guarantee on the utility of LMM.

**Theorem 3** (Utility guarantee)**.** *Pick any* $\eta \in (0, 1)$. *Suppose* $D \in \mathcal{X}^n$ *satisfies the* $(\ell^*, \gamma^*)$-*margin condition with*

$$\gamma^* = \frac{21}{n\alpha}\ln\frac{3}{\eta} + T^{(\ell^*)}.$$

*Then with probability at least* $1 - \eta$, $I := \text{LMM}(\alpha, \delta, D)$ *satisfies*

$$f(I, D) \geq f^{(1)}(D) - \frac{6\ln(2\ell^*/\eta)}{n\alpha}.$$

*(Above, $T^{(\ell^*)}$ is as defined in Algorithm 1.)*

*Remark* 1. Fix some $\alpha, \delta \in (0, 1)$. Theorem 3 states that if the dataset $D$ satisfies the $(\ell^*, \gamma^*)$-margin condition, for some positive integer $\ell^*$ and $\gamma^* = C \log(\ell^*/\delta)/(n\alpha)$ for some universal constant $C > 0$, then the value $f(I, D)$ of the item $I$ returned by LMM is within $O(\log(\ell^*)/(n\alpha))$ of the maximum, with high probability. There is no explicit dependence on the cardinality $K$ of the universe $\mathcal{U}$.

## 4 Illustrative Applications

We now describe applications of LMM to problems from data mining and machine learning.

### 4.1 Private Frequent Itemset Mining

Frequent Itemset Mining (FIM) is the following popular data mining problem: given the purchase lists of users (say, for an online grocery store), the goal is to find the sets of items that are purchased together most often. The work of [5] provides the first differentially private algorithms for FIM. However, as these algorithms rely on the exponential mechanism and the max-of-Laplaces mechanism, their utilities degrade with the total number of possible itemsets. Subsequent algorithms exploit other properties of itemsets or avoid directly finding the most frequent itemset [34, 27, 15, 8].

Let $\mathcal{I}$ be the set of items that can be purchased, and let $B$ be the maximum length of an user's purchase list. Let $\mathcal{U} \subseteq 2^{\mathcal{I}}$ be the family of itemsets of interest. For simplicity, we let $\mathcal{U} := \binom{\mathcal{I}}{r}$—i.e., all itemsets of size $r$—and consider the problem of picking the itemset with the (approximately) highest frequency. This is a private maximization problem where $D$ is the users' lists of purchased items, and $f(i, D)$ is the fraction of users who purchase an itemset $i \in \mathcal{U}$. Let $f_{\max}$ be the highest frequency of an itemset in $D$. Let $L$ be the total number of itemsets with non-zero frequency, so $L \leq n\binom{B}{r}$, which is $\ll |\mathcal{I}|^r$ whenever $B \ll |\mathcal{I}|$. Applying LMM gives the following guarantee.

**Corollary 1.** *Suppose we use* LMM$(\alpha, \delta, \cdot)$ *on the FIM problem above. Then there exists a constant $C > 0$ such that the following holds. If $f_{\max} \geq C \cdot \log(L/\delta)/(n\alpha)$, then with probability $\geq 1 - \delta$, the frequency of the itemset $I_{\text{LMM}}$ output by* LMM *is*

$$f(I_{\text{LMM}}, D) \geq f_{\max} - O\left(\frac{\log(L/\delta)}{n\alpha}\right).$$

In contrast, the itemset $I_{\text{EM}}$ returned by the exponential mechanism is only guaranteed to satisfy

$$f(I_{\text{EM}}, D) \geq f_{\max} - O\left(\frac{r \log(|\mathcal{I}|/\delta)}{n\alpha}\right),$$

which is significantly worse than Corollary 1 whenever $L \ll |\mathcal{I}|^r$ (as is typically the case). Second, to ensure differential privacy by running the exponential mechanism, one needs *a priori* knowledge of the set $\mathcal{U}$ (and thus the universe of items $\mathcal{I}$) independently of the observed data; otherwise the process will not be end-to-end differentially private. In contrast, our algorithm does not need to know $\mathcal{I}$ in order to provide end-to-end differential privacy. Finally, unlike [31], our algorithm does not require a gap between the top two itemset frequencies.

### 4.2 Private PAC Learning

We now consider private PAC learning with a finite hypothesis class $\mathcal{H}$ with bounded VC dimension $d$ [25]. Here, the dataset $D$ consists of $n$ labeled training examples drawn iid from a fixed distribution. The error $\text{err}(h)$ of a hypothesis $h \in \mathcal{H}$ is the probability that it misclassifies a random example drawn from the same distribution. The goal is to return a hypothesis $h \in \mathcal{H}$ with error as low as possible. A standard procedure that has been well-studied in the literature simply returns the minimizer $\hat{h} \in \mathcal{H}$ of the empirical error $\widehat{\text{err}}(h, D)$ computed on the training data $D$, but this does not guarantee (approximate) differential privacy. The work of [25] instead uses the exponential mechanism to select a hypothesis $h_{\text{EM}} \in \mathcal{H}$. With probability $\geq 1 - \delta_0$,

$$\text{err}(h_{\text{EM}}) \leq \min_{h \in \mathcal{H}} \text{err}(h) + O\left(\sqrt{\frac{d \log(n/\delta_0)}{n}} + \frac{\log |\mathcal{H}| + \log(1/\delta_0)}{\alpha n}\right). \tag{2}$$

The dependence on $\log |\mathcal{H}|$ is improved to $d \log |\Sigma|$ by [7] when the data entries come from a finite set $\Sigma$. The subsequent work of [4] introduces the notion of representation dimension, and shows how it relates to differentially private learning in the discrete and finite case, and [3] provides improved convergence bounds with approximate differential privacy that exploit the structure of some specific hypothesis classes. For the case of infinite hypothesis classes and continuous data distributions, [10] shows that distribution-free private PAC learning is not generally possible, but distribution-dependent learning can be achieved under certain conditions.

We provide a sample complexity bound of a rather different character compared to previous work. Our bound only relies on uniform convergence properties of $\mathcal{H}$, and can be significantly tighter than the bounds from [25] when the number of hypotheses with error close to $\min_{h \in \mathcal{H}} \mathrm{err}(h)$ is small. Indeed, the bounds are a private analogue of the *shell bounds* of [26], which characterize the structure of the hypothesis class as a function of the properties of a decomposition based on hypotheses' error rates. In many situation, these bounds are significantly tighter than those that do not involve the error distributions.

Following [26], we divide the hypothesis class $\mathcal{H}$ into $R = O(\sqrt{n/(d \log n)})$ shells; the $t$-th shell $\mathcal{H}(t)$ is defined by

$$\mathcal{H}(t) := \left\{ h \in \mathcal{H} : \mathrm{err}(h) \leq \min_{h' \in \mathcal{H}} \mathrm{err}(h') + C_0 t \sqrt{\frac{d \log(n/\delta_0)}{n}} \right\}.$$

Above, $C_0 > 0$ is the constant from uniform convergence bounds—i.e., $C_0$ is the smallest $c > 0$ such that for all $h \in \mathcal{H}$, with probability $\geq 1 - \delta_0$, we have $|\widehat{\mathrm{err}}(h, D) - \mathrm{err}(h)| \leq c\sqrt{d \log(n/\delta_0)/n}$. Observe that $\mathcal{H}(t + 1) \subseteq \mathcal{H}(t)$; and moreover, with probability $\geq 1 - \delta_0$, all $h \in \mathcal{H}(t)$ have $\widehat{\mathrm{err}}(h, D) \leq \min_{h' \in \mathcal{H}} \mathrm{err}(h') + C_0 \cdot (t + 1)\sqrt{d \log(n/\delta_0)/n}$.

Let $t^*(n)$ as the smallest integer $t \in \mathbb{N}$ such that

$$\frac{\log(|\mathcal{H}(t + 1)|) + \log(1/\delta)}{t} \leq \frac{C_0 \alpha \sqrt{dn \log n}}{C}$$

where $C > 0$ is the constant from Remark 1. Then, with probability $\geq 1 - \delta_0$, the dataset $D$ with $f = 1 - \widehat{\mathrm{err}}$ satisfies the $(\ell, \gamma)$-margin condition, with $\ell = |\mathcal{H}(t^*(n) + 1)|$ and $\gamma = C \log(|\mathcal{H}(t^*(n) + 1)|/\delta)/(\alpha n)$. Therefore, we have the following guarantee for applying LMM to this problem.

**Corollary 2.** *Suppose we use* LMM$(\alpha, \delta, \cdot)$ *on the learning problem above (with* $\mathcal{U} = \mathcal{H}$ *and* $f = 1 - \widehat{\mathrm{err}}$*). Then, with probability* $\geq 1 - \delta_0 - \delta$*, the hypothesis* $h_{\mathrm{LMM}}$ *returned by* LMM *satisfies*

$$\mathrm{err}(h_{\mathrm{LMM}}) \leq \min_{h \in \mathcal{H}} \mathrm{err}(h) + O\left(\sqrt{\frac{d \log(n/\delta_0)}{n}} + \frac{\log(|\mathcal{H}(t^*(n) + 1)|/\delta)}{\alpha n}\right).$$

The dependence on $\log |\mathcal{H}|$ from (2) is replaced here by $\log(|\mathcal{H}(t^*(n) + 1)|/\delta)$, which can be vastly smaller, as discussed in [26].

## 5 Additional Related Work

There has been a large amount of work on differential privacy for a wide range of statistical and machine learning tasks over the last decade [6, 30, 13, 21, 33, 24, 1]; for overviews, see [18] and [29]. In particular, algorithms for the private maximization problem (and variants) have been used as subroutines in many applications; examples include PAC learning [25], principle component analysis [14], performance validation [12], and multiple hypothesis testing [32].

A separation between pure and approximate differential privacy has been shown in several previous works [19, 31, 3]. The first approximate differentially private algorithm that achieves a separation is the Propose-Test-Release (PTR) framework [19]. Given a function, PTR determines an upper bound on its local sensitivity at the input dataset through a search procedure; noise proportional to this upper bound is then added to the actual function value. We note that the PTR framework does not directly apply to our setting as the sensitivity is not generally defined for a discrete universe.

In the context of private PAC learning, the work of [3] gives the first separation between pure and approximate differential privacy. In addition to using the algorithm from [31], they devise two

additional algorithmic techniques: a concave maximization procedure for learning intervals, and an algorithm for the private maximization problem under the $\ell$-bounded growth condition discussed in Section 2.2. The first algorithm is specific to their problem and does not appear to apply to general private maximization problems. The second algorithm has a sample complexity bound of $n = O(\log(\ell)/\alpha)$ when the function $f$ satisfies the $\ell$-bounded growth condition.

Lower bounds for approximate differential privacy have been shown by [7, 16, 11, 9], and the proof of our Theorem 1 borrows some techniques from [11].

## 6 Conclusion and Future Work

In this paper, we have presented the first general and range-independent algorithm for approximate differentially private maximization. The algorithm automatically adapts to the available large margin properties of the sensitive dataset, and reverts to worst-case guarantees when such properties are lacking. We have illustrated the applicability of the algorithm in two fundamental problems from data mining and machine learning; in future work, we plan to study other applications where range-independence is a substantial boon.

**Acknowledgments.** We thank an anonymous reviewer for suggesting the simpler variant of LMM based on the exponential mechanism. (The original version of LMM used a *max of truncated exponentials* mechanism, which gives the same guarantees up to constant factors.) This work was supported in part by the NIH under U54 HL108460 and the NSF under IIS 1253942.

## Footnotes

[1]Our notion of margins here is different from the usual notion of margins from statistical learning that underlies linear prediction methods like support vector machines and boosting. In fact, our notion is more closely related to the shell decomposition bounds of [26], which we discuss in Section 4.2.

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
