[Supplementary Material]

**Algorithm 2** $\mathsf{M}(\alpha, D)$

---

**input** Privacy parameter $\alpha > 0$, database $D \in \mathcal{X}^n$.
**output** Max estimate $m \in \mathbb{R}$.
 1: Draw $Z \sim \mathrm{Lap}(1/\alpha)$.
 2: **return** $f^{(1)}(D) + Z/n$.

---

**Algorithm 3** $\mathsf{S}(\alpha, m, \theta_1, \theta_2, \ldots, \theta_{K-1}, D)$

---

**input** Privacy parameter $\alpha > 0$, max estimate $m \in \mathbb{R}$, thresholds $\theta_1, \theta_2, \ldots, \theta_{K-1} \in \mathbb{R}$, database $D \in \mathcal{X}^n$.
**output** Rank $r \in \{1, 2, \ldots, K\}$.
 1: Draw $G \sim \mathrm{Lap}(2/\alpha)$ and $Z_1, Z_2, \ldots, Z_{K-1} \overset{\text{iid}}{\sim} \mathrm{Lap}(4/\alpha)$
 2: **for** $r = 1, 2, \ldots, K-1$ **do**
 3:    **if** $m - f^{(r+1)}(D) > (Z_r + G)/n + \theta_r$ **then**
 4:       **return** $r$.
 5:    **end if**
 6: **end for**
 7: **return** $K$.

---

# A  Appendix

## A.1  Privacy Analysis

In this section, we present the proof of Theorem 2. We rely on composition results for approximate differential privacy to analyze the three parts of Algorithm 1:

- Differential privacy of releasing $m$ after Step 3.
- Differential privacy of releasing $\ell$ after Step 12.
- Approximate differential privacy of releasing $I$ after Step 15.

We make this explicit by encapsulating these parts in Algorithm 2 ($\mathsf{M}$), Algorithm 3 ($\mathsf{S}$), and Algorithm 4 ($\mathsf{A}$), so we can write Algorithm 1 as follows (after the definitions of $T^{(r)}$ and $t^{(r)}$):

1. $m := \mathsf{M}(\alpha/3, D)$.
2. $\ell := \mathsf{S}(\alpha/3, m, T^{(1)}, T^{(2)}, \ldots, T^{(K-1)}, D)$.
3. $I := \mathsf{A}(\alpha/3, \ell, D)$.

### A.1.1  max **Estimation**

The first part of Algorithm 1 is a standard application of the Laplace mechanism; it is detailed in Algorithm 2.

**Lemma 1** ([17]). $\mathsf{M}(\alpha, \cdot)$ *is $\alpha$-differentially private.*

**Lemma 2.** *With probability at least $1 - \delta$,*

$$\mathsf{M}(\alpha, D) \le f^{(1)}(D) + \frac{1}{n\alpha} \ln \frac{1}{2\delta}.$$

*Proof.* This follows from the tail properties of the Laplace distribution. □

### A.1.2  Certifying the Margin Condition

The second part of Algorithm 1 is an application of the "sparse vector technique" to certify the margin condition; it is detailed in Algorithm 3.

**Lemma 3.** *For any $m, \theta_1, \theta_2, \ldots, \theta_{K-1} \in \mathbb{R}$, $\mathsf{S}(\alpha, m, \theta_1, \theta_2, \ldots, \theta_{K-1}, \cdot)$ is $\alpha$-differentially private.*

*Proof.* This is an application of the sparse vector technique from [22] that halts as soon as the first "query" is answered positively. We give the privacy analysis for completeness. For clarity, we suppress the dependence of $\mathsf{S}$ on all inputs except $D$, and define $F^{(r+1)} := m - f^{(r+1)} - \theta_r$, which inherits the $(1/n)$-Lipschitz property from $f^{(r+1)}$.

Pick any neighboring datasets $D$ and $D'$, and pick any $\ell \in \{1, 2, \ldots, K\}$. We use the notation $\Pr_{|G}(\cdot)$ for conditional probabilities where the value of $G$ is fixed, so $\Pr(\cdot) = \mathbb{E}(\Pr_{|G}(\cdot))$, where the expectation is taken with respect to $G$. Observe that

$$\Pr_{|G}(\mathsf{S}(D) = \ell) = \Pr_{|G}(\mathsf{S}(D) \le \ell | \mathsf{S}(D) > \ell - 1) \prod_{r=1}^{\ell-1} \Pr_{|G}(\mathsf{S}(D) > r | \mathsf{S}(D) > r - 1). \quad (3)$$

From the definition of $\mathsf{S}$ and $F^{(r+1)}$,

$$\Pr_{|G}(\mathsf{S}(D) > r | \mathsf{S}(D) > r - 1) = \Pr_{|G}\left(F^{(r+1)}(D) \le \frac{Z_r + G}{n}\right) \quad \forall r \in \{1, 2, \ldots, \ell-1\},$$

and

$$\Pr_{|G}(\mathsf{S}(D) \le \ell | \mathsf{S}(D) > \ell - 1) = \Pr_{|G}\left(F^{(\ell+1)}(D) > \frac{Z_\ell + G}{n}\right).$$

Write $\boldsymbol{Z}_{1:\ell-1} := (Z_1, Z_2, \ldots, Z_{\ell-1})$, and define for any $g \in \mathbb{R}$,

$$\mathcal{Z}_g(D) := \left\{ \boldsymbol{z} \in \mathbb{R}^{\ell-1} : F^{(r+1)}(D) \le \frac{z_r + g}{n} \quad \forall r \in \{1, 2, \ldots, \ell-1\} \right\},$$

so that

$$\prod_{r=1}^{\ell-1} \Pr_{|G}(\mathsf{S}(D) > r | \mathsf{S}(D) > r - 1) = \prod_{r=1}^{\ell-1} \Pr_{|G}\left(F^{(r+1)}(D) \le \frac{Z_r + G}{n}\right)$$
$$= \Pr_{|G}\left(\boldsymbol{Z}_{1:\ell-1} \in \mathcal{Z}_G(D)\right).$$

Hence, substituting into (3), we have

$$\Pr_{|G}(\mathsf{S}(D) = \ell) = \Pr_{|G}\left(F^{(\ell+1)}(D) > \frac{Z_\ell + G}{n}\right) \Pr_{|G}(\boldsymbol{Z}_{1:\ell-1} \in \mathcal{Z}_G(D)).$$

Letting $p$ denote the density of $G$, we have the following chain of inequalities:

$$\Pr(\mathsf{S}(D) = \ell) = \mathbb{E}(\Pr_{|G}(\mathsf{S}(D) = \ell))$$
$$= \int_{-\infty}^{\infty} \Pr_{|G}\left(F^{(\ell+1)}(D) > \frac{Z_\ell + g}{n}\right) \Pr_{|G}(\boldsymbol{Z}_{1:\ell-1} \in \mathcal{Z}_g(D)) p(g) dg$$
$$\le \exp(\alpha/2) \int_{-\infty}^{\infty} \Pr_{|G}\left(F^{(\ell+1)}(D) > \frac{Z_\ell + g}{n}\right) \Pr_{|G}(\boldsymbol{Z}_{1:\ell-1} \in \mathcal{Z}_g(D)) p(g+1) dg \quad (4)$$
$$= \exp(\alpha/2) \int_{-\infty}^{\infty} \Pr_{|G}\left(F^{(\ell+1)}(D) > \frac{Z_\ell + g - 1}{n}\right) \Pr_{|G}(\boldsymbol{Z}_{1:\ell-1} \in \mathcal{Z}_{g-1}(D)) p(g) dg$$
$$\le \exp(\alpha/2) \int_{-\infty}^{\infty} \Pr_{|G}\left(F^{(\ell+1)}(D) > \frac{Z_\ell + g - 1}{n}\right) \Pr_{|G}(\boldsymbol{Z}_{1:\ell-1} \in \mathcal{Z}_g(D')) p(g) dg \quad (5)$$
$$\le \exp(\alpha) \int_{-\infty}^{\infty} \Pr_{|G}\left(F^{(\ell+1)}(D') > \frac{Z_\ell + g}{n}\right) \Pr_{|G}(\boldsymbol{Z}_{1:\ell-1} \in \mathcal{Z}_g(D')) p(g) dg \quad (6)$$
$$= \exp(\alpha) \Pr(\mathsf{S}(D') = \ell).$$

To prove (4), we use the fact $p(g) \le \exp(\alpha/2)p(g+1)$ since $p$ is the Laplace density with scale parameter $\alpha/2$. To prove (5), observe that for all $r \in \{1, 2, \ldots, \ell-1\}$, the $(1/n)$-Lipschitz property of $F^{(r+1)}$ implies

$$F^{(r+1)}(D) \le \frac{Z_r + g - 1}{n} \implies F^{(r+1)}(D') \le \frac{Z_r + g}{n}.$$

---
**Algorithm 4** $\mathsf{A}(\alpha, \ell, D)$
---
**input** Privacy parameter $\alpha > 0$, number of items $\ell > 0$, database $D \in \mathcal{X}^n$.
**output** Item $I \in \mathcal{U}$.
  1: Let $\mathcal{U}_\ell$ be the set of $\ell$ items in $\mathcal{U}$ with highest $f(i, D)$ value, ties broken arbitrarily.
  2: Draw $I \sim \boldsymbol{p}$ where $p_i \propto \mathbb{1}\{i \in \mathcal{U}_\ell\} \exp(n\alpha f(i, D)/2)$.
  3: **return** $I$.

---

This, in turn, implies $\mathcal{Z}_{g-1}(D) \subseteq \mathcal{Z}_g(D')$, so (5) follows. To prove (6), we use the following. Observe that

$$F^{(\ell+1)}(D) > \frac{Z_\ell + g - 1}{n} \implies F^{(\ell+1)}(D') > \frac{Z_\ell + g - 2}{n}$$

by the $(1/n)$-Lipschitz property of $F^{(\ell+1)}$. Therefore

$$\mathrm{Pr}_{|G}\left(F^{(\ell+1)}(D) > \frac{Z_\ell + g - 1}{n}\right) \leq \mathrm{Pr}_{|G}\left(F^{(\ell+1)}(D') > \frac{Z_\ell + g - 2}{n}\right)$$

$$\leq \exp(\alpha/2)\mathrm{Pr}_{|G}\left(F^{(\ell+1)}(D') > \frac{Z_\ell + g}{n}\right)$$

where we use the fact that $Z_\ell \sim \mathrm{Lap}(\alpha/4)$ for the last step, so (6) follows. $\qquad\square$

**Lemma 4.** *With probability at least $1 - \delta$, if $\mathsf{S}(\alpha, m, \theta_1, \theta_2, \ldots, \theta_{K-1}, D) = r$ then*

$$m - f^{(r+1)}(D) > \theta_r - \frac{2}{n\alpha} \ln \frac{1}{\delta} - \frac{4}{n\alpha} \ln \frac{r(r+1)}{\delta}.$$

*Proof.* Using the tail bound for the Laplace distribution,

$$\Pr\left(G < -\frac{2}{\alpha} \ln \frac{1}{\delta}\right) \leq \frac{\delta}{2}$$

and

$$\Pr\left(Z_r < -\frac{4}{\alpha} \ln \frac{r(r+1)}{\delta}\right) \leq \frac{\delta}{2r(r+1)}$$

for each $r \in \{1, 2, \ldots, K-1\}$. Therefore, by a union bound, with probability at least $1 - \delta$,

$$G \geq -\frac{2}{\alpha} \ln \frac{1}{\delta} \quad \text{and} \quad Z_r \geq -\frac{4}{\alpha} \ln \frac{r(r+1)}{\delta} \; \forall r \in \{1, 2, \ldots, K-1\}.$$

The claim follows. $\qquad\square$

### A.1.3 Restricted Exponential Mechanism

The third part of Algorithm 1 uses the exponential mechanism on the top $\ell$ items to select one of these items; it is detailed in Algorithm 4.

**Lemma 5.** *Assume $D$ satisfies the $(\ell, \gamma)$-margin condition with*

$$\gamma \geq \frac{2}{n}\left(1 + \frac{\ln(\ell/\beta)}{\alpha}\right).$$

*Then for any neighbor $D' \in \mathcal{X}^n$ of $D$, and any $S \subseteq \mathcal{U}$,*

$$\Pr(\mathsf{A}(\alpha, D) \in S) \leq \exp(\alpha) \cdot \Pr(\mathsf{A}(\alpha, D') \in S) + \beta.$$

*Proof.* For any $r \in \{1, 2, \ldots, K\}$ and dataset $\tilde{D} \in \mathcal{X}^n$, let $H_{\tilde{D}} \subseteq \mathcal{U}$ denote the $r$ items of highest $f(\cdot, \tilde{D})$ value (ties broken arbitrarily). (In Algorithm 4, we have $\mathcal{U}_\ell = H_D$.) It suffices to show that

$$\Pr(\mathsf{A}(\alpha, \ell, D') = i) \leq \max\{\Pr(\mathsf{A}(\alpha, \ell, D) = i) \exp(\alpha), \beta/\ell\}, \quad \forall i \in H_{D'}.$$

This is because $\Pr(\mathsf{A}(\alpha, \ell, D') \notin H_{D'}) = 0$ and $|H_{D'}| = \ell$.

Fix any $i \in H_{D'}$. Because $f(j, \cdot)$ is $(1/n)$-Lipschitz for every $j \in \mathcal{U}$, so is $f^{(r)}(\cdot)$ for every $r \in [K]$. Therefore

$$\sum_{r=1}^{\ell} \exp\left(\frac{n\alpha}{2} f^{(r)}(D')\right) \geq \sum_{r=1}^{\ell} \exp\left(\frac{n\alpha}{2} f^{(r)}(D)\right) \exp(-\alpha/2).$$

Also by the $(1/n)$-Lipschitz property,

$$\exp\left(\frac{n\alpha}{2} f(i, D')\right) \leq \exp\left(\frac{n\alpha}{2} f(i, D)\right) \exp(\alpha/2).$$

Therefore, combining the two displayed equations above gives

$$\Pr(\mathsf{A}(\alpha, \ell, D') = i) = \frac{\exp\left(\frac{n\alpha}{2} f(i, D')\right)}{\sum_{r=1}^{\ell} \exp\left(\frac{n\alpha}{2} f^{(r)}(D')\right)} \leq \frac{\exp\left(\frac{n\alpha}{2} f(i, D)\right)}{\sum_{r=1}^{\ell} \exp\left(\frac{n\alpha}{2} f^{(r)}(D)\right)} \exp(\alpha). \quad (7)$$

If $i \in H_D$, then (7) reads

$$\Pr(\mathsf{A}(\alpha, \ell, D') = i) \leq \Pr(\mathsf{A}(\alpha, \ell, D) = i) \exp(\alpha).$$

If $i \notin H_D$, then the assumption that $D$ satisfies the $(\ell, \gamma)$-margin condition implies

$$f(i, D) \leq f^{(1)}(D) - \gamma;$$

so combining the above inequality with (7), as well as the assumption $\gamma \geq (2/n)(1 + \ln(\ell/\beta)/\alpha)$, gives

$$\Pr(\mathsf{A}(\alpha, \ell, D') = i) \leq \frac{\exp\left(\frac{n\alpha}{2}\left(f^{(1)}(D) - \gamma\right)\right)}{\exp\left(\frac{n\alpha}{2} f^{(1)}(D)\right)} \exp(\alpha) \leq \beta/\ell. \qquad \square$$

### A.1.4  Privacy of Algorithm 1

For clarity, we suppress the privacy parameter inputs to the algorithms. By standard composition results for differential privacy [17], Lemma 1, and Lemma 3, the release of $\mathsf{M}(D)$ and $\mathsf{S}(\mathsf{M}(D), D)$ is $(2\alpha/3)$-differentially private. Define the shorthand $\mathsf{MS}(D) := (\mathsf{M}(D), \mathsf{S}(\mathsf{M}(D), D))$, and let $\mu_D$ denote the corresponding probability measure over the range of $\mathsf{MS}(D)$.

For a dataset $D \in \mathcal{X}^n$, let $\mathcal{V}_D$ be set of $(\tilde{m}, \tilde{\ell})$ pairs (i.e., possible outputs of $\mathsf{MS}$) such that

$$\tilde{m} \leq f^{(1)}(D) + \frac{3}{n\alpha} \ln \frac{3}{2\delta} \quad \text{and} \quad \tilde{m} - f^{(\tilde{\ell}+1)}(D) > T^{(\tilde{\ell})} - \frac{12}{n\alpha} \ln \frac{3\tilde{\ell}(\tilde{\ell}+1)}{\delta} - \frac{6}{n\alpha} \ln \frac{3}{\delta}.$$

If $(m, \ell) \in \mathcal{V}_D$, then the values of $T^{(\ell)}$ and $t^{(\ell)}$ certify that $D$ satisfies the $(\ell, t^{(\ell)})$-margin condition. Lemma 2 and Lemma 4 imply that

$$\mu_D(\mathcal{V}_D) \geq 1 - \frac{2\delta}{3}.$$

Also, observe that if $\beta := \delta \exp(-2\alpha/3)/3$, then

$$t^{(\ell)} = \frac{2}{n}\left(1 + \frac{\ln(\ell/\beta)}{\alpha/3}\right).$$

Therefore, for any neighbor $D' \in \mathcal{X}^n$ of $D$, and any $S \subseteq \mathcal{U}$,

$$\Pr(\textsc{lmm}(D) \in S) = \int \Pr(\mathsf{A}(\ell, D) \in S \,|\, \mathsf{MS}(D) = (m, \ell)) d\mu_D$$

$$\leq \int_{\mathcal{V}_D} \Pr(\mathsf{A}(\ell, D) \in S \,|\, \mathsf{MS}(D) = (m, \ell)) d\mu_D + \frac{2\delta}{3}$$

$$\leq \int_{\mathcal{V}_D} \left(e^{\alpha/3} \Pr(\mathsf{A}(\ell, D') \in S \,|\, \mathsf{MS}(D) = (m, \ell)) + \beta\right) e^{2\alpha/3} d\mu_{D'} + \frac{2\delta}{3}$$

$$= \int_{\mathcal{V}_D} \left(e^{\alpha/3} \Pr(\mathsf{A}(\ell, D') \in S \,|\, \mathsf{MS}(D') = (m, \ell)) + \frac{\delta e^{-2\alpha/3}}{3}\right) e^{2\alpha/3} d\mu_{D'} + \frac{2\delta}{3}$$

$$\leq \int \left(e^{\alpha/3} \Pr(\mathsf{A}(\ell, D') \in S \,|\, \mathsf{MS}(D') = (m, \ell)) + \frac{\delta e^{-2\alpha/3}}{3}\right) e^{2\alpha/3} d\mu_{D'} + \frac{2\delta}{3}$$

$$= e^{\alpha} \Pr(\textsc{lmm}(D') \in S) + \delta.$$

Above, the second inequality follows from Lemma 5 and the $(2\alpha/3)$-differential privacy of $\mathsf{MS}$. $\quad \square$

## A.2 Utility Analysis

*Proof of Theorem 3.* Using tail bounds for the Laplace distribution, it follows that with probability at least $1 - \eta/2$,

$$Z \geq -\frac{3}{\alpha}\ln\frac{3}{\eta}, \quad G \leq \frac{6}{\alpha}\ln\frac{3}{\eta}, \quad Z_{\ell^*} \leq \frac{12}{\alpha}\ln\frac{3}{\eta}.$$

In this event, the assumption that $D$ satisfies the $(\ell^*, \gamma^*)$-margin condition implies that

$$\left(f^{(1)}(D) + Z/n\right) - f^{(\ell^*+1)}(D) > (Z_{\ell^*} + G)/n + T^{(\ell^*)},$$

so the while-loop terminates with $\ell \leq \ell^*$. Also, the probability distribution $p$ in Step 14 of Algorithm 1 assigns probability mass at most $\eta/2$ to the set of items $i$ with

$$f(i, D) \leq f^{(1)}(D) - \frac{6\ln(2\ell/\eta)}{n\alpha}.$$

Therefore, by a union bound, the item $I$ returned by Algorithm 1 satisfies

$$f(I, D) > f^{(1)}(D) - \frac{6\ln(2\ell^*/\eta)}{n\alpha}$$

with probability at least $1 - \eta$. $\qquad\square$

## A.3 Proofs of Lower Bounds

*Proof of Theorem 1.* We construct the private maximization problem as follows. Let the domain $\mathcal{X} := 2^{\mathcal{U}}$ (subsets of items), and define $f : \mathcal{U} \times \mathcal{X}^n \to \mathbb{R}$ by

$$f(i, D) := \frac{1}{n}\sum_{s=1}^{n}\mathbb{1}\{i \in D_s\}.$$

In other words, the function $f(i, \cdot)$ is the fraction of entries containing $i$. It is easy to see that $f(i, \cdot)$ is $(1/n)$-Lipschitz for all $i \in \mathcal{U}$.

Let $m := \min\{n/2, \log((\ell-1)/2)/\alpha\}$. We define a collection of $\ell$ datasets $D^1, D^2, \ldots, D^\ell \in \mathcal{X}^n$ with the following properties:

1. For each $i$, the first $n/2$ entries of $D^i$ are equal to $[\ell] := \{1, 2, \ldots, \ell\}$, the next $n/2 - m$ are equal of $D^i$ are equal to $\emptyset$, and the last $m$ entries of $D^i$ are equal to $\{i\}$. Therefore

$$f(j, D^i) = \begin{cases} 0 & \text{if } j \notin [\ell], \\ \frac{1}{2} & \text{if } j \in [\ell] \setminus \{i\}, \\ \frac{1}{2} + \frac{m}{n} & \text{if } j = i, \end{cases}$$

so $f(i, D^i) = f^{(1)}(D^i)$ and $D^i$ satisfies the $(\ell, m/n)$-margin condition.

2. For each $i \neq j$, the datasets $D^i$ and $D^j$ differ only in (the last) $m$ entries.

Let $\mathcal{A}$ be $(\alpha, \delta)$-approximate differentially private. Assume for sake of contradiction that

$$\Pr\left(f(\mathcal{A}(D^i), D^i) > f^{(1)}(D^i) - \frac{m}{n}\right) \geq \frac{1}{2}$$

for all $i \in [\ell]$. Since only $i$ satisfies $f(i, D^i) > f^{(1)}(D^i) - m/n$, this is the same as $\Pr(\mathcal{A}(D^i) = i) \geq 1/2$ for all $i \in [\ell]$. This then implies the following chain of inequalities leading to a contradiction:

$$\frac{1}{2} > \Pr(\mathcal{A}(D^i) \neq i)$$

$$\geq \sum_{j \in [\ell]\setminus\{i\}}\Pr(\mathcal{A}(D^i) = j)$$

$$\geq \sum_{j \in [\ell]\setminus\{i\}}e^{-\alpha m}\Pr(\mathcal{A}(D^j) = j) - \frac{\delta}{1 - e^{-\alpha}}$$

$$\geq (\ell-1)\left(\frac{e^{-\alpha m}}{2} - \frac{\delta}{1 - e^{-\alpha}}\right) \geq \frac{1}{2}.$$

The first inequality above is by assumption; the third inequality follows from Lemma 6; the fourth inequality again uses the assumption; and the final inequality follows by the definition of $m$ and the condition on $\delta$. Since a contradiction is reached, there must exist some $i \in [\ell]$ such that $\Pr(f(\mathcal{A}(D^i), D^i) > f^{(1)}(D^i) - m/n) < 1/2$. □

**Lemma 6** ([11]). *Let $D$ and $D'$ be any two datasets that differ in at most $k$ entries, and let $\mathcal{A}$ be any $(\alpha, \delta)$-approximate differentially private algorithm with range $\mathcal{S}$. Then, for any $S \subseteq \mathcal{S}$,*

$$\Pr(\mathcal{A}(D) \in S) \geq e^{-k\alpha} \Pr(\mathcal{A}(D') \in S) - \frac{\delta}{1 - e^{-\alpha}}.$$