[Reviews · NeurIPS 2014]

Submitted by Assigned_Reviewer_3

This paper looks at differentially private algorithms for a generic maximization problem ("private argmax" might be a good name). Given a collection of K of items, and a data set D of n individuals, and a "score" function f that assigns each item i a data-based score f(i;D), the goal is to find an item i with approximately maximal score, while preserving differential privacy.

This "private argmax" has proven to be a fundamental problem in the theory of private data analysis. It was first formulated by McSherry and Talwar (2007), who proposed the "exponential mechanism" to solve it. Because of its generality, that mechanism has been a central tool for designing many differentially private algorithms. The performance of that algorithm is optimal in the worst case, even among "approximately DP" algorithms, but it has been known for some time that there are various simple conditions under which it can be improved substantially.

The authors give a private argmax algorithm which is differentially private (actually, a commonly-studied relaxation called "approximate DP") and which does better than the exponential mechanism for a large class of inputs. The authors call these instances "large margin" instances -- essentially, they assume that there are not too many items with near-optimal margin. Previously, it was known how to solve this problem when there is a large gap between the best and second-best items; this paper generalizes that technique considerably.

The authors illustrate their results by showing theoretical improvements for a two problems. The more interesting of these is PAC learning -- the authors show that their algorithms allows one to exploit "shell bounds" (essentially, conditions which imply that not too many concept classes are good for the data set with high probability).

This is a nice result that has the potential to influence future work considerably, as it provides a useful and basic algorithmic tool. It is not completely unexpected, given recent work of a similar flavor, but it takes those ideas much further. I recommend accepting the paper.

The paper also presents lower bounds showing that the parameters of the algorithm cannot be asymptotically improved. The second of these (for (alpha,delta)-DP) is a nice contribution independently of the rest of the paper.

Comments:

I think the presentation of the algorithm could be somewhat simplified.

- For any given ell, there is a simple mechanism which does well and is private *under the (ell,gamma) assumption* for gamma >= log(\ell/delta)/alpha. Namely, just run the exponential mechanism on the top \ell items. (In a nutshell, it is private since the probability that one outputs an item with frequency f_\ell is at most delta). This is somewhat simpler than the truncated exponential mechanism the authors employ.

- One can select a good ell using the sparse vector technique. This is essentially what the authors do.

Such a two-stage explanation might make the algorithm a little easier to parse.

This paper would probably be more appealing to the NIPS community with some sort of empirical evaluation, but I think it is more than strong enough as it is for acceptance.
Summary: New algorithm for a basic algorithmic problem in private data analysis, with applications to PAC learning.

Submitted by Assigned_Reviewer_16

This paper describes a new mechanism that outputs a (randomized) maximizer of f(i , D) w.r.t. discrete i \in I in a differentially private manner.
This problem can be solved by the exponential mechanism with taking f(i, D) as the utility function or max-of-Laplaces mechanism.
According to the authors analysis, without any assumption on D and f, the lower bound on the output of the mechanism is dependent on K, the size of the item universe. 
The authors' idea is to assume that there exist a certain separation in f(i, D), between the top ell choice of i and the remaining choices, which is called (\ell, \gamma)-margin. Under the condition of (\ell, \gamma)-margin, the authors describe the large-margin mechanism with guarantee of (\alpha, \delta)-differential privacy. The authors also show that the utility guaranteed by this mechanism is independent on the item universe, but is dependent on \ell.
Applications of the proposed mechanism to PAC learning and frequent item-set mining have been shown, too.

The idea seems to be novel and covers a wide range of applications, not only to ML but also to other domains.
The paper is well organized and easy to understand.

Comments:
- Inequality (1) may be opposite? It is unclear what D_j denotes.

Summary: The idea seems to be novel and covers a wide range of applications, not only to ML but also to other domains. Independency on K is an obvious merit, whereas I think some experiments that demonstrate the practicality of the proposed mechanism are still needed.

Submitted by Assigned_Reviewer_36

This paper studies the private maximization problem: given a database D, a two-variable function f(i, D) where i belongs to a set U, the goal is to find i* so that f(i*,D) approximately achieves the maximum while preserving the privacy of D. The problem contains Private Empirical Risk Minimization as a special case. Previous private maximization algorithms has one limitation, the performance deteriorates with an increasing universe size |U|. This paper shows that under a certain large margin condition, it is possible to have an algorithm that does not suffer from the limitation mentioned above.

The way I view this paper is that it generalizes the results in the COLT’13 paper *Differentially Private Feature Selection via Stability Arguments and the Robustness of Lasso*, due to Smith and Thakurta (Ref. [8]). In the COLT paper, it is assumed that there is a single maximize i* such that f(i*,D) is larger than all other f(i,D) by a certain margin. It can be shown that under this assumption there is a private algorithm whose performance does not decrease with increasing |U|.

In the current paper, the authors make a more general assumption. Let f^(1) be the largest value f(i*,D), f^(2) be the second largest value and so on. The assumption in this paper is that the f^(i) decreases fast as i grows. The authors then propose a private large margin mechanism (LMM) whose performance depends only on how fast f^(i) decreases, but not on the size of the universe |U|.

I think this paper is very interesting. The private maximization problem is general. A differentially private mechanism for this problem may have many applications in machine learning and data mining. On the other hand, I have some concerns about the results.

- The large margin mechanism (LMM) is highly inefficient. It has to calculate f(i,D) for every i in the universe U. The focus of this paper is the case that |U| is large. In fact if |U| is substantially smaller than exp(|D|), one does not need to worry about the decrease of the performance. So the result in this paper is particularly useful when |U| is at least exp(|D|). But in this case the computational cost of LMM seems prohibitive for any application of interest. Note that the algorithm of the COLT'13 paper is efficient if the non-private version is efficient.

- Is it possible to give a learning application such that one can quantify the benefits LMM provides? The authors give an example of private PAC learning, but the improvement of the performance depend on other parameters such as the size of the shell of the hypothesis space.

Other detailed comments:

The statement of Theorem 3 is problematic. It is easy to see that there can be many (l*, gamma*) satisfying the margin condition. I think you mean the smallest l*.

**********************************************************************

Comments after reading the author feedback:

The authors partially answered some of my questions. I suggest the authors discuss the computational issues in the paper and, if possible, provide a concrete example that LMM is efficient in the large margin setting.

Summary: Interesting work. But the algorithm is computationally prohibitive. It would be better to have an example to quantify the benefit of the new mechanism.
Author Feedback
Author rebuttal: We thank all the reviewers for their thoughtful feedback; we will
incorporate the suggestions in the revision.

Reviewer_36:

Our algorithm is a generalization of [8]. Our margin condition is novel,
and as shown by our lower bounds, this concept is critical for extending beyond ell=2.

Our algorithm is not significantly more computationally intensive than [8]
when there is a large margin. For example, if the
conditions of [8] hold (within constant factors), then our algorithm
is essentially the same as that of [8]. More generally, instead of requiring
the top two highest f(i, D)’s as in [8], if there is a (l,gamma)-margin, then our
algorithm needs to find the highest l+1 f(i,D)’s.

In many specific cases, our algorithm can be efficiently implemented even when
K is exponentially large in the database size. For example, in FIM,
we can use techniques from [4] and only calculate the
frequencies of item sets that actually occur in the data, as the rest
of the frequencies are zero.

Note that prior to our work, it was not even clear that general
range-independence private maximization is possible from an
information-theoretic standpoint, let alone what general mechanisms
are able to achieve it.

A case where LMM provides quantifiable benefits is the problem of
learning threshold functions over the unit interval on the real line.
Recall that distribution-free private learning of this hypothesis
class is not possible; so consider a finite cover of this hypothesis
class with respect to a non-private reference distribution. It is easy
to construct a data distribution such that the size of the cover
required to guarantee a low error is arbitrarily large, but there are
just a few hypotheses with low error.

Reviewer_16:

Indeed, the inequality in Eq. (1) should read D_j \geq i. We use D_j
to denote the value of the j-th entry in the dataset D. The items are
numbered 1 through K; the value of item i is the
fraction of dataset entries with value at least i. We will fix this
typo and and clarify the notations.

Reviewer_3:

Thanks for your detailed comments and the suggested simplifications. Indeed, we've confirmed that this variant of the exponential mechanism also works
under the margin condition, and hence will simplify the presentation.